Statins: antimicrobial resistance breakers or makers?

Ko Humphrey H.T. h.ko2@student.curtin.edu.au 1 2
Lareu Ricky R. 1 2
Dix Brett R. 1
Hughes Jeffery D. 1
1 School of Pharmacy, Faculty of Health Sciences, Curtin University , Perth , Western Australia , Australia
2 Curtin Health Innovation Research Institute (CHIRI) Biosciences Research Precinct, Curtin University , Perth , Western Australia , Australia
Flores-Valdez Mario Alberto
Electronic publication date: 2017 Oct 24
Publication date: 2017
Volume: 5
Electronic Location ID: e3952
Received 2017 May 19; Accepted 2017 Oct 2
Copyright: ©2017 Ko et al.
Copyright year: 2017
Copyright holder: Ko et al.
License: This is an open access article distributed under the terms of the Creative Commons Attribution License, which permits unrestricted use, distribution, reproduction and adaptation in any medium and for any purpose provided that it is properly attributed. For attribution, the original author(s), title, publication source (PeerJ) and either DOI or URL of the article must be cited.
License URL: https://creativecommons.org/licenses/by/4.0/

Keywords: Minimum inhibitory concentration, Statins, Antimicrobial resistance, Antibacterial mechanism, Drug repurposing

Funding: The authors did not receive external funding for this review article.

==============================
Introduction

The repurposing of non-antibiotic drugs as adjuvant antibiotics may help break antimicrobial resistance (AMR). Statins are commonly prescribed worldwide to lower cholesterol. They also possess qualities of AMR “breakers”, namely direct antibacterial activity, synergism with antibiotics, and ability to stimulate the host immune system. However, statins’ role as AMR breakers may be limited. Their current extensive use for cardiovascular protection might result in selective pressures for resistance, ironically causing statins to be AMR “makers” instead. This review examines statins’ potential as AMR breakers, probable AMR makers, and identifies knowledge gaps in a statin-bacteria-human-environment continuum. The most suitable statin for repurposing is identified, and a mechanism of antibacterial action is postulated based on structure-activity relationship analysis.

Methods

A literature search using keywords “statin” or “statins” combined with “minimum inhibitory concentration” (MIC) was performed in six databases on 7th April 2017. After screening 793 abstracts, 16 relevant studies were identified. Unrelated studies on drug interactions; antifungal or antiviral properties of statins; and antibacterial properties of mevastatin, cerivastatin, antibiotics, or natural products were excluded. Studies involving only statins currently registered for human use were included.

Results

Against Gram-positive bacteria, simvastatin generally exerted the greatest antibacterial activity (lowest MIC) compared to atorvastatin, rosuvastatin, and fluvastatin. Against Gram-negative bacteria, atorvastatin generally exhibited similar or slightly better activity compared to simvastatin, but both were more potent than rosuvastatin and fluvastatin.

Discussion

Statins may serve as AMR breakers by working synergistically with existing topical antibiotics, attenuating virulence factors, boosting human immunity, or aiding in wound healing. It is probable that statins’ mechanism of antibacterial activity involves interference of bacterial cell regulatory functions via binding and disrupting cell surface structures such as wall teichoic acids, lipoteichoic acids, lipopolysaccharides, and/or surface proteins. The widespread use of statins for cardiovascular protection may favor selective pressures or co-selection for resistance, including dysbiosis of the human gut microbiota, sublethal plasma concentrations in bacteremic patients, and statin persistence in the environment, all possibly culminating in AMR.

Conclusion

Simvastatin appears to be the most suitable statin for repurposing as a novel adjuvant antibiotic. Current evidence better supports statins as potential AMR breakers, but their role as plausible AMR makers cannot be excluded. Elucidating the mechanism of statins’ antibacterial activity is perhaps the most important knowledge gap to address as this will likely clarify statins’ role as AMR breakers or makers.

Introduction

Antimicrobial resistance (AMR) occurs when microorganisms become immune to antimicrobials via intrinsic resistance (possessing mechanisms which reduce intracellular concentrations of antimicrobials or render antimicrobials ineffective); acquired resistance (gaining resistant genes via mutation or horizontal gene transfer); or adaptive resistance (adapting to environmental stress by altering gene expressions) (Canton et al., 2013; Fernandez, Breidenstein & Hancock, 2011). Selective pressures for resistance can occur at both lethal and sublethal drug concentrations (Hughes & Andersson, 2017). When susceptible bacteria are exposed to antimicrobial concentrations within eight to ten times above the minimum inhibitory concentration (MIC), AMR may occur due to the propagation of pre-existing resistant mutant strains whilst the susceptible strains are killed (Andersson & Hughes, 2014; Canton et al., 2013; Levison & Levison, 2009). At low antibiotic concentrations (up to several hundred times below MIC), AMR proliferation may occur with the growth of multiple new resistant mutant strains due to minute reductions in the growth rate of susceptible bacteria (Andersson & Hughes, 2011; Andersson & Hughes, 2014; Kohanski, DePristo & Collins, 2010).

In addition to antibiotics, it was found that exposure of bacteria to biocides, metals, and non-antibiotic chemicals with antibacterial properties also contributed to AMR via co-selection of resistant genes (Li et al., 2016; Singer et al., 2016; Wales & Davies, 2015). Co-selection protects a bacterial strain against multiple antibiotic classes due to the selection of one gene which confers multiple resistance mechanisms (cross-resistance), or the selection of physically linked genes which collectively confer various resistance mechanisms (co-resistance) (Singer et al., 2016; Wales & Davies, 2015).

The World Health Organization has warned that with the rise of AMR, the world is moving towards a post-antibiotic era whereby if last-line antibiotics become ineffective, common infections and minor injuries may prove fatal (World Health Organization, 2016b). In response to the AMR threat, many countries have initiated a concerted “One Health” best practice approach to suppress AMR, involving optimal use of antibiotics in humans and animals (World Health Organization, 2016a). It has been suggested that AMR may be impeded by the administration of certain non-antibiotic drugs together with current antibiotic treatment (Brown, 2015).These non-antibiotic drugs may be repurposed (used to treat new conditions) to act as AMR “breakers” if they have direct antibacterial activity, synergize with antibiotics, stimulate the host immune system, or possess a combination of these properties (Brown, 2015). Antihyperlipidemic agents 3-hydroxy-3-methylglutaryl-coenzyme A (HMG-CoA) reductase inhibitors, commonly known as statins, appear to possess the mentioned properties of AMR breakers and have been poised to be repurposed as novel adjuvant antimicrobials (Hennessy et al., 2016).

Statins are one of the most commonly prescribed medicines in the world, with over 30 million people in the United States and up to 200 million people worldwide taking statins daily to lower cholesterol for primary and secondary prevention of cardiovascular diseases (Blaha & Martin, 2013). By competitively binding to HMG-CoA reductase in a dose-dependent manner, statins inhibit the rate limiting step of the mevalonate pathway, thus diminishing cholesterol production (Liao, 2005). In the process however, important isoprenoid intermediates such as geranylgeranyl pyrophosphate (GGPP) and farnesyl pyrophosphate (FPP) are also reduced, hence decreasing cell signaling proteins (e.g., Ras, Rac, and Rho) and causing multiple cholesterol-independent (pleiotropic) effects which are cardioprotective (e.g., antithrombotic, antioxidant, antiplatelet, and endothelial protection) and immunomodulatory (e.g., anti-inflammatory and neutrophil extracellular trap [NET] production) (Chow et al., 2010; Gazzerro et al., 2012; Kozarov, Padro & Badimon, 2014).

Research on statins originated with the intention of developing new antibiotics. In 1971, Professor Akira Endo searched for new antibiotics with the hypothesis that fungi may produce substances which inhibit HMG-CoA reductase, thereby killing microorganisms (Endo, 2010). The discovery of statins and their potent cholesterol-lowering abilities soon led to their clinical use in preventing cardiovascular diseases instead (Endo, 2010). In recent years however, interest returned to the inherent antimicrobial effects of statins (Jerwood & Cohen, 2008).

Although statins possess the potential to be AMR breakers (direct antibacterial activity, synergistic activity with antibiotics, and ability to stimulate the human immune system) (Brown, 2015; Hennessy et al., 2016), they are currently extensively used to treat hypercholesterolemia (a non-antimicrobial purpose). Prolonged exposure of bacterial populations to drugs with antibacterial properties may expedite the death of susceptible bacteria, resulting in subsequent dominance of resistant bacteria, regardless of the exposure being in humans, animals, or the environment (Canton et al., 2013). The problem is likely to be compounded with recent guidelines recommending the initiation of statins in adults aged 40 to 75 years with one or more cardiovascular risk factors (US Preventive Services Task Force, 2016), and evidence that the benefits of statins for cardiovascular protection far outweigh their side effects (Collins et al., 2016).

This review examines the potential of statins as AMR breakers, which albeit promising, could be limited by antibacterial resistance acquired via selective pressures and co-selection, ironically culminating in statins contributing as AMR “makers” instead. Statins’ potential roles as AMR breakers, AMR makers, and knowledge gaps were reviewed as a statin-bacteria-human-environment continuum. From MIC data available in literature, the susceptibility of various bacteria to individual statins may be ascertained to reveal the most suitable statin for repurposing as a novel adjuvant antimicrobial. In addition, by comparing chemical structures of statins with antibacterial activity against statins without antibacterial activity, a mechanism of antibacterial action for statins was postulated.

Figure 1 Flow chart summarizing the literature search process performed in six databases on 7th April 2017.

CINAHL, Cumulative Index to Nursing and Allied Health Literature.

Methods

Literature search

The keywords “statin” or “statins” were combined with “minimum inhibitory concentration” to identify studies which reported MIC values of statins when tested against specific bacterial strains. “Minimum inhibitory concentration” was used as a keyword instead of a general term “antibacterial effect” because MIC values allow quantitative comparisons of antibacterial potency between individual statins (Dafale et al., 2016). Moreover, exposure of susceptible bacteria to antibacterial drug concentrations ranging from within eight to ten times above MIC to several hundred times below MIC may contribute to selective pressures for resistance (Andersson & Hughes, 2011; Levison & Levison, 2009). The search was performed by the primary investigator (HK) in six databases on 7th April 2017, namely the Cumulative Index to Nursing and Allied Health Literature (CINAHL), Cochrane Library, Embase, PubMed, Google Scholar, and Web of Science (Fig. 1).

Studies selection

Screening the titles and abstracts of the initial 793 results identified from the keywords, 756 studies were excluded because they covered unrelated topics such as drug interactions; antifungal or antiviral properties of statins; and antibacterial properties of mevastatin, cerivastatin, antibiotics, or natural products. Although antibacterial effects of mevastatin and cerivastatin have been studied (Hennessy et al., 2016), they are not currently used clinically and were therefore omitted in this review (Tobert, 2003). Only antibacterial properties of atorvastatin (ATV), fluvastatin (FLV), lovastatin (LVS), pitavastatin (PTV), pravastatin (PRV), rosuvastatin (RSV), and simvastatin (SMV) were considered relevant for this review as these are currently registered drugs for lowering cholesterol in humans, thus likely to affect the statin-bacteria-human-environment continuum.

Upon reviewing the full text of the remaining 37 studies, 21 studies were further excluded as they contained duplicate information; studied the effects of statins on infected cells instead of direct bacterial exposure; or tested the combined effects of statins and antibiotics without reporting the MIC of statins alone. The resultant 16 pertinent studies consisted of a thesis (Alshammari, 2016), a letter with unpublished MIC data (Bjorkhem-Bergman, Lindh & Bergman, 2011), a Turkish study with relevant data in its English abstract (Coban et al., 2010), a patent application (Quivey, 2014), a review article with information from a reference in press (Ting, Whitaker & Albandar, 2016), and 11 in vitro studies (Bergman et al., 2011; Emani, Gunjiganur & Mehta, 2014; Graziano et al., 2015; Jerwood & Cohen, 2008; Masadeh et al., 2012; Matzneller, Manafi & Zeitlinger, 2011; Radwan & Ezzat, 2012; Sarabhai et al., 2015; Thangamani et al., 2015; Wang et al., 2016; Welsh, Kruger & Faoagali, 2009). No new relevant studies were found after scrutinizing the references of these 16 studies. The relevance of references was reviewed by all the researchers.

Data extraction

From the 16 selected studies, the MIC values of statins against various Gram-positive and Gram-negative bacteria were compiled in Tables 1 and 2 respectively. The dilution methods for Alshammari (2016), Bergman et al. (2011), Quivey (2014), Welsh, Kruger & Faoagali (2009), and Ting, Whitaker & Albandar (2016) were described in the respective studies. All other studies were tested according to the broth microdilution method stipulated by the Clinical and Laboratory Standards Institute (CLSI), formerly known as National Committee for Clinical Laboratory Standards (NCCLS). The solvent types and solvent concentrations for water insoluble statins (ATV, LVS, PTV, and SMV) were listed wherever available, because different solvents or solvent concentrations may affect the MIC values (Matzneller, Manafi & Zeitlinger, 2011).

Table 1 Compiled antimicrobial susceptibility results of statins against various Gram-positive bacteria reported in literaturea.

Bacteria type and strainb	Solvent/Brothc	Statin (MIC in µg/mL) d	Reference	
		ATV	FLV	LVS	PTV	PRV	RSV	SMV		
Bacillus species	
Isolates	Methanol 1:2 dilution (range from 50% to 0.78%)	43.75 ± 17.12	Not tested	Not tested	Not tested	Not tested	Not tested	Not tested	Radwan & Ezzat (2012)	
Bacillus anthracis	
AMES35, UM23	Unknown solvent and %	Not tested	Not tested	Not tested	Not tested	Not tested	Not tested	16	Thangamani et al. (2015)	
Enterococcus faecalis	
Unknown strain	Ethanol 1%	Not tested	Not tested	Not tested	Not tested	Not tested	Not tested	64	Quivey (2014)	
Enterococcus faecalis (Vancomycin-resistant)	
ATCC 51299	DMSO Unknown %	166.67 ± 72.16	Not tested	Not tested	Not tested	Not tested	500 ± 0.00	104.17 ± 36.08	Masadeh et al. (2012)	
ATCC 51299	Unknown solvent and %	Not tested	Not tested	Not tested	Not tested	Not tested	Not tested	32	Thangamani et al. (2015)	
ATCC 51299	Ethanol 6.25%	250	Not tested	Not tested	Not tested	Not tested	100	Not tested	Welsh, Kruger & Faoagali (2009)	
SF24413, SF28073	Unknown solvent and %	Not tested	Not tested	Not tested	Not tested	Not tested	Not tested	32	Thangamani et al. (2015)	
Isolates	DMSO Unknown %	216.67 ± 32.27	Not tested	Not tested	Not tested	Not tested	500.00 ± 0.00	291.67 ± 39.53	Masadeh et al. (2012)	
Isolates	Unknown solvent and %	>128	Not tested	Not tested	Not tested	Not tested	Not tested	>128	Coban et al. (2010)	
Enterococcus faecalis (Vancomycin-sensitive)	
ATCC 7080, ATCC 14506	Unknown solvent and %	Not tested	Not tested	Not tested	Not tested	Not tested	Not tested	32	Thangamani et al. (2015)	
ATCC 19433	DMSO Unknown %	83.33 ± 36.08	Not tested	Not tested	Not tested	Not tested	333.33 ± 144.33	52.08 ± 18.04	Masadeh et al. (2012)	
ATCC 29212	Unknown solvent and %	>128	Not tested	Not tested	Not tested	Not tested	Not tested	64	Coban et al. (2010)	
ATCC 29212	Ethanol 6.25%	250	Not tested	Not tested	Not tested	Not tested	100	Not tested	Welsh, Kruger & Faoagali (2009)	
ATCC 29212	DMSO 2.5%	>250	Not tested	Not tested	Not tested	>250	Not tested	>250	Graziano et al. (2015)	
ATCC 49532, ATCC 49533, HH22, MMH594, SF24397	Unknown solvent and %	Not tested	Not tested	Not tested	Not tested	Not tested	Not tested	32	Thangamani et al. (2015)	
Isolates	DMSO Unknown %	95.83 ± 22.09	Not tested	Not tested	Not tested	Not tested	333.33 ± 0.00	291.67 ± 39.53	Masadeh et al. (2012)	
Isolates	Unknown solvent and %	>128	Not tested	Not tested	Not tested	Not tested	Not tested	>128	Coban et al. (2010)	
Enterococcus faecium	
Unknown strain	Ethanol 1%	Not tested	Not tested	Not tested	Not tested	Not tested	Not tested	64	Quivey (2014)	
Enterococcus faecium (Vancomycin-resistant)	
ATCC 700221, E0120, ERV102	Unknown solvent and %	Not tested	Not tested	Not tested	Not tested	Not tested	Not tested	32	Thangamani et al. (2015)	
Isolates	Unknown solvent and %	>128	Not tested	Not tested	Not tested	Not tested	Not tested	>128	Coban et al. (2010)	
Enterococcus faecium (Vancomycin-sensitive)	
ATCC 6569, E1162	Unknown solvent and %	Not tested	Not tested	Not tested	Not tested	Not tested	Not tested	32	Thangamani et al. (2015)	
Isolates	Unknown solvent and %	>128	Not tested	Not tested	Not tested	Not tested	Not tested	>128	Coban et al. (2010)	
Lactobacillus casei										
Unknown strain	Not specified	Not tested	Not tested	Not tested	Not tested	Not tested	Not tested	7.8	Ting, Whitaker & Albandar (2016)	
Listeria monocytogenes										
ATCC 13932, ATCC 19111, ATCC 19112, ATCC 19114, F4244, J0161	Unknown solvent and %	Not tested	Not tested	Not tested	Not tested	Not tested	Not tested	32	Thangamani et al. (2015)	
Staphylococci (Methicillin-resistant coagulase negative, MRCoNS)	
Isolates	Unknown solvent and %	>128	Not tested	Not tested	Not tested	Not tested	Not tested	>128	Coban et al. (2010)	
Staphylococcus aureus	
Unknown strain	Ethanol 1%	Not tested	Not tested	Not tested	Not tested	Not tested	Not tested	64	Quivey (2014)	
Staphylococcus aureus(Methicillin-resistant, MRSA)	
ATCC 14458, ATCC 33591, ATCC 43300	DMSO 2.5%	>250	Not tested	Not tested	Not tested	>250	Not tested	31.25	Graziano et al. (2015)	
ATCC 43300	DMSO Unknown %	83.33 ± 36.08	Not tested	Not tested	Not tested	Not tested	500 ± 0.00	166.67 ± 72.16	Masadeh et al. (2012)	
ATCC 43300	Unknown solvent and %	>128	Not tested	Not tested	Not tested	Not tested	Not tested	>128	Coban et al. (2010)	
ATCC 43300	Unknown solvent and %	>1,024	>1,024	>1,024	>1,024	>1,024	>1,024	32	Thangamani et al. (2015)	
ATCC 49476	Ethanol 6.25%	250	Not tested	Not tested	Not tested	Not tested	100	Not tested	Welsh, Kruger & Faoagali (2009)	
ATCC BAA-44, NRS70, NRS71, NRS108, NRS119, NRS123	Unknown solvent and %	Not tested	Not tested	Not tested	Not tested	Not tested	Not tested	32	Thangamani et al. (2015)	
NRS100, NRS194	Unknown solvent and %	Not tested	Not tested	Not tested	Not tested	Not tested	Not tested	64	Thangamani et al. (2015)	
USA100, USA200, USA300, USA400, USA500, USA700, USA800, USA1000, USA1100	Unknown solvent and %	Not tested	Not tested	Not tested	Not tested	Not tested	Not tested	32	Thangamani et al. (2015)	
Isolates	DMSO Unknown %	108.33 ± 27.36	Not tested	Not tested	Not tested	Not tested	500.00 ± 0.00	116.67 ± 30.19	Masadeh et al. (2012)	
Isolates	Unknown solvent and %	>128	Not tested	Not tested	Not tested	Not tested	Not tested	>128	Coban et al. (2010)	
Isolates	Methanol 1:2 dilution (range from 50% to 0.2%)	Not tested	>200 (mean)	Not tested	Not tested	Not tested	Not tested	74.9 (mean)	Jerwood & Cohen (2008)	
Isolates	Methanol 1:2 dilution (range from 50% to 0.78%)	37.5 ± 13.98	Not tested	Not tested	Not tested	Not tested	Not tested	Not tested	Radwan & Ezzat (2012)	
Staphylococcus aureus(Methicillin-sensitive, MSSA)	
ATCC 6538	DMSO 2.5%	>250	Not tested	Not tested	Not tested	>250	Not tested	31.25	Graziano et al. (2015)	
ATCC 6538	Unknown solvent and %	Not tested	Not tested	Not tested	Not tested	Not tested	Not tested	32	Thangamani et al. (2015)	
ATCC 25213	DMSO Unknown %	41.67 ± 18.04	Not tested	Not tested	Not tested	Not tested	208.33 ± 72.16	26.04 ± 9.02	Masadeh et al. (2012)	
ATCC 25923	Unknown solvent and %	>128	Not tested	Not tested	Not tested	Not tested	Not tested	64	Coban et al. (2010)	
ATCC 25923	Ethanol 6.25%	250	Not tested	Not tested	Not tested	Not tested	100	Not tested	Welsh, Kruger & Faoagali (2009)	
ATCC 29213	DMSO 0.5%	Not tested	Not tested	Not tested	Not tested	Not tested	Not tested	62.5	Wang et al. (2016)	
ATCC 29213	Unknown solvent and %	>128	Not tested	Not tested	Not tested	Not tested	Not tested	32	Coban et al. (2010)	
ATCC 29213	DMSO 2.5%	>250	Not tested	Not tested	Not tested	>250	Not tested	15.65	Graziano et al. (2015)	
ATCC 29213	Various solvents and %	>250 (Ethanol 5%)	500	>500 (DMSO 5%)	Not tested	>500	>500	31 (Methanol 100%); 500 (Methanol 5%); 500 (SMV sodium)	Matzneller, Manafi & Zeitlinger (2011)	
RN4220, NRS72, NRS77, NRS846, NRS860	Unknown solvent and %	Not tested	Not tested	Not tested	Not tested	Not tested	Not tested	32	Thangamani et al. (2015)	
Isolates	Unknown solvent and %	>128	Not tested	Not tested	Not tested	Not tested	Not tested	>128	Coban et al. (2010)	
Isolates	DMSO Unknown %	52.08 ± 11.04	Not tested	Not tested	Not tested	Not tested	341.67 ± 20.84	60.42 ± 12.76	Masadeh et al. (2012)	
Isolates	Methanol 1:2 dilution (range from 50% to 0.2%)	Not tested	>200 (mean)	Not tested	Not tested	Not tested	Not tested	29.2 (mean)	Jerwood & Cohen (2008)	
Isolates	DMSO 2.5%	>250	Not tested	Not tested	Not tested	>250	Not tested	31.25	Graziano et al. (2015)	
Staphylococcus aureus(Vancomycin-intermediate, VISA)	
NRS1, NRS19, NRS37	Unknown solvent and %	Not tested	Not tested	Not tested	Not tested	Not tested	Not tested	32	Thangamani et al. (2015)	
Staphylococcus aureus(Vancomycin-resistant, VRSA)	
VRS1, VRS2, VRS3a, VRS3b, VRS4, VRS5, VRS6, VRS7, VRS8, VRS10, VRS11a, VRS11b, VRS12, VRS13	Unknown solvent and %	Not tested	Not tested	Not tested	Not tested	Not tested	Not tested	32	Thangamani et al. (2015)	
VRS9	Unknown solvent and %	Not tested	Not tested	Not tested	Not tested	Not tested	Not tested	64	Thangamani et al. (2015)	
Staphylococcus epidermidis										
ATCC 12228	DMSO Unknown %	20.83 ± 9.02	Not tested	Not tested	Not tested	Not tested	166.67 ± 72.16	26.04 ± 9.02	Masadeh et al. (2012)	
NRS101	Unknown solvent and %	Not tested	Not tested	Not tested	Not tested	Not tested	Not tested	32	Thangamani et al. (2015)	
Isolates	DMSO Unknown %	19.78 ± 4.94	Not tested	Not tested	Not tested	Not tested	233.33 ± 39.52	35.41 ± 4.94	Masadeh et al. (2012)	
Streptococcus anginosus										
Unknown strain	Not specified	Not tested	Not tested	Not tested	Not tested	Not tested	Not tested	7.8	Ting, Whitaker & Albandar (2016)	
Streptococcus mutans										
ATCC 25175	DMSO 1:2 dilution (range from 50% to 0.2%)	100	Not tested	Not tested	Not tested	200	100	15.6	Alshammari (2016)	
UA159	Ethanol 1%	Not tested	Not tested	Not tested	Not tested	Not tested	Not tested	16	Quivey (2014)	
Unknown strain	Not specified	Not tested	Not tested	Not tested	Not tested	Not tested	Not tested	15.6	Ting, Whitaker & Albandar (2016)	
Streptococcus pneumoniae										
51916, 70677	Unknown solvent and %	Not tested	Not tested	Not tested	Not tested	Not tested	Not tested	64	Thangamani et al. (2015)	
ATCC BAA-334	DMSO 2.5%	Not tested	>100	Not tested	Not tested	>100	Not tested	15.6	Bergman et al. (2011)	
Unknown ATCC strain	DMSO Unknown %	104.17 ± 36.08	Not tested	Not tested	Not tested	Not tested	333.33 ± 144.33	166.67 ± 72.16	Masadeh et al. (2012)	
Isolates	DMSO Unknown %	229.17 ± 60.38	Not tested	Not tested	Not tested	Not tested	416.67 ± 0.00	291.67 ± 39.53	Masadeh et al. (2012)	
Unknown strain	Unknown solvent and %	Not tested	Not tested	Not tested	Not tested	Not tested	Not tested	15	Bjorkhem-Bergman et al. (2011)	
Streptococcus pyogenes										
ATCC 19615	DMSO Unknown %	83.33 ± 36.08	Not tested	Not tested	Not tested	Not tested	166.67 ± 72.16	62.5 ± 0.00	Masadeh et al. (2012)	
Isolates	DMSO Unknown %	133.33 ± 19.76	Not tested	Not tested	Not tested	Not tested	275.00 ± 72.17	145.83 ± 32.27	Masadeh et al. (2012)	
Streptococcus salivarius										
ATCC 2593	DMSO 1:2 dilution (range from 50% to 0.2%)	100	Not tested	Not tested	Not tested	200	100	7.8	Alshammari (2016)	
Unknown strain	Not specified	Not tested	Not tested	Not tested	Not tested	Not tested	Not tested	7.8	Ting, Whitaker & Albandar (2016)	
Streptococcus sanguinis (Streptococcus sanguis)	
ATCC 10556	DMSO 1:2 dilution (range from 50% to 0.2%)	100	Not tested	Not tested	Not tested	200	100	15.6	Alshammari (2016)	
Unknown strain	Not specified	Not tested	Not tested	Not tested	Not tested	Not tested	Not tested	15.6	Ting, Whitaker & Albandar (2016)	
Notes.

a The dilution methods for Alshammari (2016), Bergman et al. (2011), Quivey (2014), Welsh, Kruger & Faoagali (2009), and Ting, Whitaker & Albandar (2016) were described in the respective studies. All other studies were tested according to the broth microdilution method stipulated by the Clinical and Laboratory Standards Institute (CLSI), formerly known as National Committee for Clinical Laboratory Standards (NCCLS).

b ATCC, American Type Culture Collection.

c All studies were tested with Mueller Hinton broth unless specified. Solvent types and solvent concentrations used for water insoluble statins (ATV, LVS, PTV, and SMV) were listed as reported in the various references. DMSO, dimethyl sulfoxide.

d ATV, atorvastatin; FLV, fluvastatin; LVS, lovastatin; MIC, minimum inhibitory concentration; PRV, pravastatin; PTV, pitavastatin; RSV, rosuvastatin; SMV, simvastatin.

Table 2 Compiled antimicrobial susceptibility results of statins against various Gram-negative bacteria reported in literaturea.

Bacteria type and strainb	Solvent/Brothc	Statin (MIC in µg/mL) d	Reference	
		ATV	FLV	LVS	PTV	PRV	RSV	SMV		
Acinetobacter baumannii										
ATCC 17978	DMSO Unknown %	15.62 ± 0.00	Not tested	Not tested	Not tested	Not tested	333.33 ± 144.33	104.17 ± 36.08	Masadeh et al. (2012)	
ATCC BAA747, ATCC BAA1605, ATCC BAA19606	Unknown solvent and %	Not tested	Not tested	Not tested	Not tested	Not tested	Not tested	>256	Thangamani et al. (2015)	
Isolates	DMSO Unknown %	21.87 ± 4.94	Not tested	Not tested	Not tested	Not tested	300.00 ± 79.05	32.29 ± 6.38	Masadeh et al. (2012)	
Isolates	Unknown solvent and %	>128	Not tested	Not tested	Not tested	Not tested	Not tested	>128	Coban et al. (2010)	
Aggregatibacter actinomycetemcomitans	
Unknown ATCC strain	DMSO 1% stock, Brain heart infusion broth	Not tested	Not tested	Not tested	Not tested	Not tested	Not tested	<1	Emani, Gunjiganur & Mehta (2014)	
Unknown strain	Not specified	Not tested	Not tested	Not tested	Not tested	Not tested	Not tested	3.95	Ting, Whitaker & Albandar (2016)	
Citrobacter freundii										
ATCC 8090	DMSO Unknown %	83.33 ± 36.08	Not tested	Not tested	Not tested	Not tested	166.67 ± 72.16	52.08 ± 18.04	Masadeh et al. (2012)	
Isolates	DMSO Unknown %	108.33 ± 27.36	Not tested	Not tested	Not tested	Not tested	333.33 ± 79.06	133.33 ± 39.58	Masadeh et al. (2012)	
Enterobacter aerogenes										
ATCC 29751	DMSO Unknown %	15.62 ± 0.00	Not tested	Not tested	Not tested	Not tested	104.17 ± 36.08	26.04 ± 9.02	Masadeh et al. (2012)	
Isolates	DMSO Unknown %	19.78 ± 4.94	Not tested	Not tested	Not tested	Not tested	183.33 ± 0.00	33.33 ± 4.94	Masadeh et al. (2012)	
Enterobacter cloacae										
ATCC 13047	DMSO Unknown %	41.67 ± 18.04	Not tested	Not tested	Not tested	Not tested	166.67 ± 72.16	62.5 ± 0.00	Masadeh et al. (2012)	
Isolates	DMSO Unknown %	113.54 ± 27.06	Not tested	Not tested	Not tested	Not tested	316.67 ± 64.55	143.75 ± 36.97	Masadeh et al. (2012)	
Escherichia coli										
1411, SM1411ΔacrAB	Unknown solvent and %	Not tested	Not tested	Not tested	Not tested	Not tested	Not tested	>256	Thangamani et al. (2015)	
ATCC 10536, ATCC 25922	DMSO 2.5%	>250	Not tested	Not tested	Not tested	>250	Not tested	>250	Graziano et al. (2015)	
ATCC 25922	Various solvents and %	>250 (Ethanol 5%)	500	>500 (DMSO 5%)	Not tested	>500	>500	>500 (Methanol 100% and 5%)	Matzneller, Manafi & Zeitlinger (2011)	
ATCC 25922	Ethanol 6.25%	250	Not tested	Not tested	Not tested	Not tested	100	Not tested	Welsh, Kruger & Faoagali (2009)	
ATCC 35218	DMSO Unknown %	26.04 ± 9.02	Not tested	Not tested	Not tested	Not tested	104.17 ± 36.08	52.08 ± 18.04	Masadeh et al. (2012)	
ATCC 35218	Unknown solvent and %	>128	Not tested	Not tested	Not tested	Not tested	Not tested	>128	Coban et al. (2010)	
Isolates	DMSO Unknown %	100.00 ± 33.75	Not tested	Not tested	Not tested	Not tested	125.00 ± 16.14	112.5 ± 30.19	Masadeh et al. (2012)	
Isolates	Unknown solvent and %	>128	Not tested	Not tested	Not tested	Not tested	Not tested	>128	Coban et al. (2010)	
Isolates	Methanol 1:2 dilution (range from 50% to 0.78%)	75 ± 27.95	Not tested	Not tested	Not tested	Not tested	Not tested	Not tested	Radwan & Ezzat (2012)	
Escherichia coliO157:H7										
ATCC 35150, ATCC 700728	Unknown solvent and %	Not tested	Not tested	Not tested	Not tested	Not tested	Not tested	>256	Thangamani et al. (2015)	
Haemophilus influenzae										
ATCC 29247	DMSO Unknown %	83.33 ± 36.08	Not tested	Not tested	Not tested	Not tested	166.67 ± 72.16	52.08 ± 18.04	Masadeh et al. (2012)	
Isolates	DMSO Unknown %	104.17 ± 36.08	Not tested	Not tested	Not tested	Not tested	366.67 ± 0.00	145.83 ± 32.27	Masadeh et al. (2012)	
Isolates	DMSO 2.5%	Not tested	>100	Not tested	Not tested	>100	Not tested	>250	Bergman et al. (2011)	
Klebsiella species										
Not specified	Ethanol 1%	Not tested	Not tested	Not tested	Not tested	Not tested	Not tested	64	Quivey (2014)	
Klebsiella pneumoniae										
ATCC 13883	DMSO Unknown %	166.67 ± 72.16	Not tested	Not tested	Not tested	Not tested	333.33 ± 144.33	166.67 ± 72.16	Masadeh et al. (2012)	
ATCC 700603	Unknown solvent and %	>128	Not tested	Not tested	Not tested	Not tested	Not tested	>128	Coban et al. (2010)	
ATCC BAA-1705, ATCC BAA-2146	Unknown solvent and %	Not tested	Not tested	Not tested	Not tested	Not tested	Not tested	>256	Thangamani et al. (2015)	
Isolates	DMSO Unknown %	216.67 ± 51.03	Not tested	Not tested	Not tested	Not tested	258.33 ± 64.55	241.67 ± 60.38	Masadeh et al. (2012)	
Isolates	Unknown solvent and %	>128	Not tested	Not tested	Not tested	Not tested	Not tested	>128	Coban et al. (2010)	
Moraxella catarrhalis										
Isolates	DMSO 2.5%	Not tested	>100	Not tested	Not tested	>100	Not tested	15.6	Bergman et al. (2011)	
Porphyromonas gingivalis										
ATCC 33277	DMSO 1% stock, Brain heart infusion broth	Not tested	Not tested	Not tested	Not tested	Not tested	Not tested	2	Emani, Gunjiganur & Mehta (2014)	
Proteus mirabilis										
ATCC 12459	DMSO Unknown %	62.5 ± 0.00	Not tested	Not tested	Not tested	Not tested	250 ± 0.00	166.67 ± 72.16	Masadeh et al. (2012)	
Isolates	DMSO Unknown %	127.08 ± 25.51	Not tested	Not tested	Not tested	Not tested	191.67 ± 32.27	158.33 ± 32.27	Masadeh et al. (2012)	
Isolates	Methanol 1:2 dilution (range from 50% to 0.78%)	125 ± 0.00	Not tested	Not tested	Not tested	Not tested	Not tested	Not tested	Radwan & Ezzat (2012)	
Pseudomonas aeruginosa										
ATCC 9027	DMSO Unknown %	83.33 ± 36.08	Not tested	Not tested	Not tested	Not tested	166.67 ± 72.16	166.67 ± 72.16	Masadeh et al. (2012)	
ATCC 9027, ATCC 9721, ATCC 10145	Unknown solvent and %	Not tested	Not tested	Not tested	Not tested	Not tested	Not tested	>256	Thangamani et al. (2015)	
ATCC 15442	Unknown solvent and %	>1,024	>1,024	>1,024	>1,024	>1,024	>1,024	>1,024	Thangamani et al. (2015)	
ATCC 25619	DMSO 2.5%	>250	Not tested	Not tested	Not tested	>250	Not tested	>250	Graziano et al. (2015)	
ATCC 25619, ATCC 27853	Unknown solvent and %	Not tested	Not tested	Not tested	Not tested	Not tested	Not tested	>256	Thangamani et al. (2015)	
ATCC 27853	DMSO 2.5%	>250	Not tested	Not tested	Not tested	>250	Not tested	>250	Graziano et al. (2015)	
ATCC 27853	Various solvents and %	>250 (Ethanol 5%)	500	>500 (DMSO 5%)	Not tested	>500	>500	>500 (Methanol 100% and 5%)	Matzneller, Manafi & Zeitlinger (2011)	
ATCC 27853	Ethanol 6.25%	250	Not tested	Not tested	Not tested	Not tested	100	Not tested	Welsh, Kruger & Faoagali (2009)	
ATCC 35032, ATCC BAA-1744	Unknown solvent and %	Not tested	Not tested	Not tested	Not tested	Not tested	Not tested	>256	Thangamani et al. (2015)	
PAO1	DMSO 2% stock, Lysogeny Broth	625	Not tested	Not tested	Not tested	Not tested	625	Not tested	Sarabhai et al. (2015)	
Isolates	DMSO Unknown %	95.83 ± 22.09	Not tested	Not tested	Not tested	Not tested	291.67 ± 39.53	120.83 ± 32.27	Masadeh et al. (2012)	
Isolates	Unknown solvent and %	>128	Not tested	Not tested	Not tested	Not tested	Not tested	>128	Coban et al. (2010)	
Unknown strain	Ethanol 1%	Not tested	Not tested	Not tested	Not tested	Not tested	Not tested	>256	Quivey (2014)	
Salmonella Typhimurium										
ATCC 700720	Unknown solvent and %	Not tested	Not tested	Not tested	Not tested	Not tested	Not tested	>256	Thangamani et al. (2015)	
Notes.

a The dilution methods for Bergman et al. (2011), Quivey (2014), Welsh, Kruger & Faoagali (2009), and Ting, Whitaker & Albandar (2016) were described in the respective studies. All other studies were tested according to the broth microdilution method stipulated by the Clinical and Laboratory Standards Institute (CLSI), formerly known as National Committee for Clinical Laboratory Standards (NCCLS).

b ATCC, American Type Culture Collection.

c All studies were tested with Mueller Hinton broth unless specified. Solvent types and solvent concentrations used for water insoluble statins (ATV, LVS, PTV, and SMV) were listed as reported in the various references. DMSO, dimethyl sulfoxide.

d ATV, atorvastatin; FLV, fluvastatin; LVS, lovastatin; MIC, minimum inhibitory concentration; PRV, pravastatin; PTV, pitavastatin; RSV, rosuvastatin; SMV, simvastatin.

Results

Antibacterial activity of statins against Gram-positive bacteria

Statins exhibited antibacterial activity against a wide spectrum of Gram-positive bacteria including oral microbiota (Staphylococcus epidermidis, Streptococcus anginosus, Streptococcus mutans, Streptococcus pneumoniae, Streptococcus pyogenes, Streptococcus salivarius, and Streptococcus sanguinis, formerly known as Streptococcus sanguis); gut microbiota (Enterococcus faecalis, Enterococcus faecium, Lactobacillus casei, and methicillin-susceptible Staphylococcus aureus [MSSA]); drug-resistant bacteria (vancomycin-resistant Enterococci [VRE], methicillin-resistant S. aureus [MRSA], vancomycin-intermediate S. aureus [VISA], and vancomycin-resistant S. aureus [VRSA]); and environmental bacteria (Bacillus anthracis and Listeria monocytogenes) (Table 1).

The antibacterial activity of SMV was found to be generally the most potent (lowest MIC) compared to ATV and RSV, especially against Enterococci (MIC[SMV ] ≈ 32 to 292 µg/mL, MIC[ATV ] ≈ 83 to >250 µg/mL, MIC[RSV ] ≈ 100 to 500 µg/mL); Staphylococci (MIC[SMV ] ≈ 16 to 167 µg/mL, MIC[ATV ] ≈ 20 to >1,024 µg/mL, MIC[RSV ] ≈ 100 to >1,024 µg/mL); and Streptococci (MIC[SMV ] ≈ 7.8 to 292 µg/mL, MIC[ATV ] ≈ 83 to 229 µg/mL, MIC[RSV ] ≈ 100 to 417 µg/mL). FLV exhibited relatively weak antibacterial activity against Staphylococci (MIC[FLV ] ranged from >200 to >1,024 µg/mL) and Streptococci (MIC[FLV ] > 100 µg/mL).

SMV has been the most widely studied, with researchers examining bacteria which were not tested against other statins such as B. anthracis (MIC[SMV ] = 16 µg/mL), L. casei (MIC[SMV ] = 7.8 µg/mL), and L. monocytogenes (MIC[SMV ] = 32 µg/mL). Few studies have been performed on the other statins, but one study did compare the antibacterial effects of all seven registered statins (ATV, FLV, LVS, PTV, PRV, RSV, and SMV) against MRSA and found that only SMV exhibited antibacterial activity (MIC[SMV ] = 32 µg/mL), while all the other six statins did not (MIC > 1,024 µg/mL) (Thangamani et al., 2015).

Antibacterial activity of statins against Gram-negative bacteria

From Table 2, statins also displayed varying antibacterial activity against a range of Gram-negative bacteria, including oral microbiota (Aggregatibacter actinomycetemcomitans and Porphyromonas gingivalis); nasopharyngeal microbiota (Haemophilus influenzae and Moraxella catarrhalis); gut microbiota (Citrobacter freundii, Enterobacter aerogenes, Enterobacter cloacae, Escherichia coli, Klebsiella pneumoniae, and Proteus mirabilis); and environmental bacteria (Acinetobacter baumannii, Pseudomonas aeruginosa, and Salmonella Typhimurium).

In general, ATV demonstrated similar or slightly better antibacterial activity compared to SMV and both were more potent than RSV against A. baumannii (MIC[ATV ] ≈ 16 to >128 µg/mL, MIC[SMV ] ≈ 32 to >256 µg/mL, MIC[RSV ] ≈300 to 333 µg/mL) and E. coli (MIC[ATV ] ≈ 26 to >250 µg/mL, MIC[SMV ] ≈ 52 to >500 µg/mL, MIC[RSV ] ≈ 100 to >500 µg/mL). FLV exerted relatively weak antibacterial activity against E. coli (MIC[FLV ] = 500 µg/mL) and P. aeruginosa (MIC[FLV ] = 500 to >1,024 µg/mL). One study evaluated the antibacterial effects of all seven registered statins against P. aeruginosa but did not find any antibacterial activity (MIC > 1,024 µg/mL) (Thangamani et al., 2015).

Variations in MIC results amongst different studies

A two-fold difference in MIC, defined as the lowest antimicrobial concentration that completely inhibits microbial growth, is generally accepted (Turnidge & Paterson, 2007). However, greater differences have been reported in some cases amongst various researchers determining the MICs of statins. For example in Table 1 when SMV was tested against a reference American Type Culture Collection (ATCC) MRSA strain (ATCC 43300), the highest MIC[SMV ] (≈167 µg/mL) and lowest MIC[SMV ] (≈31 µg/mL) differed by about five-fold (Graziano et al., 2015; Masadeh et al., 2012). Variations in MIC results of a statin against the same bacterial strain between different studies could be attributed to diversity in materials and methods employed, especially if materials were obtained from different manufacturers. Slight deviations in environmental conditions during manufacture, storage, or transport may affect drug or media purity which consequently influences MIC results.

Protocols may not specify every minute detail. General instructions for water insoluble solvents allowed investigators to use various types of solvents and solvent concentrations of their choice, which may result in different MIC results (Matzneller, Manafi & Zeitlinger, 2011). Most of the studies in Tables 1 and 2 utilized the CLSI broth microdilution method protocol, which recommends an incubation time of 16 to 20 h for bacteria such as S. aureus, but does not specify if microtiter plates should be subjected to continuous shaking during incubation (Clinical and Laboratory Standards Institute, 2012). A window of 4 h may result in different MIC results between readings taken at 16 h compared with 20 h of incubation. Some researchers may choose to subject the plates to shaking during incubation to facilitate exposure of bacteria to the drug or reduce biofilm formation under static growth conditions. However, continuous shaking during incubation may cause more colonies to grow, affecting MIC results (Liu et al., 2015; Shanholtzer et al., 1984). The CLSI protocol also stipulates that the MIC should be discerned as absence of turbidity with the unaided eye (Clinical and Laboratory Standards Institute, 2012). This may lead to subjective results, depending on the ability of individuals to detect minute disparities in turbidity.

In view of the multiple factors hampering reproduction of results, it may be more meaningful to compare absolute quantitative results (e.g., MIC) within studies performed by the same researchers, whilst qualitative results or trends (e.g., spectrum of antibacterial efficacy) could be analyzed between studies by different researchers.

Discussion

The positive factors which promote the use of statins as novel adjuvant antibiotics for infections (statins as AMR breakers), the negative factors whereby acquired antibacterial resistance against statins could culminate in AMR (statins as AMR makers), and knowledge gaps are summarized in Fig. 2 and elaborated as follows.

Figure 2 Potential of statins as repurposed novel adjuvant antibiotics for infections in the statin-bacteria-human-environment continuum.

(+) refers to factors leading to potentially positive outcomes, whereby statins co-administered with antibiotics may impede AMR (AMR breakers). (−) refers to factors leading to potentially negative outcomes, whereby statin use may favor selective pressures or co-selection for resistance and culminate in AMR (AMR makers). (?) refers to further research required to bridge knowledge gap. AMR, antimicrobial resistance; MIC, minimum inhibitory concentration; NET, neutrophil extracellular trap.

AMR breaker: intrinsic antibacterial activity

The MIC values in Tables 1 and 2 provide in vitro evidence of individual statins’ inherent antibacterial effects against various Gram-positive and Gram-negative bacteria gleaned from literature thus far. SMV has been the most widely studied and demonstrated antibacterial activity against different types of microbiota (oral, gut, and nasopharyngeal) and environmental bacteria (Tables 1 and 2). SMV also exerted antibacterial effects against Gram-positive drug resistant bacteria such as MRSA, VISA, VRE, and VRSA (Table 1). Therefore, SMV may prove to be an effective antibiotic adjuvant, but in vivo studies are required to confirm its clinical antibacterial efficacy.

Knowledge gap: contribution of statins as AMR makers via selective pressures or co-selection

Despite evidence of statins’ intrinsic antibacterial effects, the life span of statins as novel adjuvant antibiotics serving as AMR breakers may be limited due to the widespread use of statins for non-antibiotic purposes (cardiovascular protection). Such extensive usage exposes susceptible bacteria in humans and the environment to varying concentrations of statins, favoring selective pressures for antibacterial resistance. The possible scenarios and repercussions of exposing susceptible bacterial strains to low (up to several hundred times below MIC) and high (within eight to ten times above MIC) statin concentrations are discussed later in this review. Emergence of AMR due to selective pressures are difficult to predict due to variable influences present in humans, animals, and the environment (Hughes & Andersson, 2017). However, it is certain that the development of AMR occurs naturally in bacteria when exposed to antimicrobials (Blair et al., 2015).

Antibiotics, biocides, metals, and non-antibiotic chemicals with antibacterial properties may also induce resistance to multiple antibiotic classes via co-selection (Singer et al., 2016; Wales & Davies, 2015). Bacteria may develop multidrug resistance via inheriting genes conferring various resistance mechanisms such as reduced cell permeability to antibiotics, increased efflux of antibiotics, modification of antibiotic targets, or direct inactivation of antibiotics (Blair et al., 2015). Co-selection occurs via cross-resistance (selection of a gene conferring multiple resistance mechanisms) or co-resistance (selection of physically linked genes which collectively confer various resistance mechanisms) (Singer et al., 2016; Wales & Davies, 2015). This is of particular concern because bacteria may inherit multidrug resistance properties in the absence of selective pressures (Wales & Davies, 2015).

To date, there is evidence that exposure of bacteria to non-antibiotic chemicals with antibacterial properties (chlorite and iodoacetic acid) may induce AMR (Li et al., 2016). Hence, there is a possibility of statins, as non-antibiotic chemicals with antibacterial properties, to similarly contribute as AMR makers, although there is currently little known evidence of such statin associations.

It was found that ATV unlikely contributed to efflux-mediated resistance in multidrug-resistant Gram-negative bacteria (Laudy, Kulinska & Tyski, 2017). As a result, statins probably contribute as AMR makers via other resistance mechanisms. More studies on statins’ mechanism of antibacterial resistance, as well as the mechanism of antibacterial activity, are required to determine and thus control the extent of statins’ plausible role as AMR makers.

Knowledge gap: mechanism of statins’ antibacterial action (Fungal origin unlikely correlates with statins’ antibacterial activity)

SMV, LVS, and PRV have been classified as Type 1 statins (derived from fungal origins and have similar chemical structures) while ATV, FLV, PTV, and RSV have been classified as Type 2 statins (synthetic compounds with chemical groups which bind more tightly with HMG-CoA reductase), as shown in Fig. 3 (Gazzerro et al., 2012). Although SMV, LVS, and PRV have similar chemical structures, SMV exhibited antibacterial properties against S. aureus but LVS and PRV do not, despite all three being of fungal origin (Thangamani et al., 2015). Moreover, ATV and RSV are synthetic compounds and not of fungal origin, but both exhibited some antibacterial activity (Masadeh et al., 2012). As such, statins’ fungal origin unlikely correlates with their antibacterial activity.

Figure 3 Chemical structures of clinically used statins and selected non-antibiotic drugs from other pharmacological classes.

(A) Type 1 statins (SMV, LVS, and PRV) are derived from fungi and have similar chemical structures. (B) Type 2 statins (ATV, FLV, PTV, and RSV) are synthetic compounds which bind more tightly with HMG-CoA reductase. (C) Selected non-antibiotic drugs from other pharmacological classes with antibacterial activity against S. aureus. The dihydroxy acid moiety (in PRV, ATV, FLV, PTV, and RSV) is required for HMG-CoA reductase inhibition, while the lactone group (in SMV and LVS) must by metabolised to the dihydroxy acid moiety before HMG-CoA reductase inhibition may occur. Drugs marked (†) possess antibacterial activity against S. aureus. Two methyl groups arranged in a tetrahedral (*) or similar trigonal pyramidal (#) molecular geometry may be important for such antibacterial activity. ATV, atorvastatin; FLV, fluvastatin; HMG-CoA, 3-hydroxy-3-methylglutaryl-coenzyme A; LVS, lovastatin; PRV, pravastatin; PTV, pitavastatin; RSV, rosuvastatin; SMV, simvastatin.

Knowledge gap: mechanism of statins’ antibacterial action (Inhibition of human or bacterial HMG-CoA reductase unlikely correlates with statins’ antibacterial activity)

When administered in humans, all statins inhibit HMG-CoA reductase in the mevalonate pathway to lower cholesterol synthesis. However, not all statins exhibit antibacterial activity (Tables 1 and 2). The presence of the dihydroxy acid moiety is required to competitively inhibit the catalytic function of HMG-CoA reductase and reduce cholesterol synthesis (Harrold, 2013). Statins with lactone groups (SMV and LVS) are prodrugs which must be metabolized to the active dihydroxy acid moiety before they may inhibit HMG-CoA reductase (Harrold, 2013). Yet SMV, being unable to directly inhibit HMG-CoA reductase, exhibits antibacterial activity against MRSA whilst PRV and PTV, being direct HMG-CoA reductase inhibitors, do not exhibit antibacterial activity (Thangamani et al., 2015).

In addition, the degree of HMG-CoA reductase inhibition corresponds directly with the cholesterol-lowering capabilities of statins (Liao & Laufs, 2005), but it does not seem commensurate with antibacterial potency. The cholesterol-lowering potency of statins has been established in the following order: PTV (most potent) >  RSV >  ATV >  SMV >  PRV >  LVS >  FLV (least potent) (Armitage, 2007). RSV is a more potent cholesterol-lowering drug compared to SMV, but SMV demonstrated greater antibacterial activity (Tables 1 and 2), indicating that antibacterial activity may not correlate with the inhibition of human HMG-CoA reductase.

Humans and some Gram-positive bacteria such as S. aureus synthesize essential isoprenoids via the mevalonate pathway (Heuston et al., 2012), whereby HMG-CoA reductase is a catalyst in the rate determining step. However, humans and bacteria have different overall HMG-CoA reductase structures. When administered in humans, statins preferentially bind to human HMG-CoA reductase (Class I) instead of bacterial HMG-CoA reductase (Class II) because the affinity of statins is about 10,000 times stronger for human HMG-CoA reductase (Friesen & Rodwell, 2004). Hence, statins are not likely to exert antibacterial effects via inhibition of bacterial HMG-CoA reductase.

Furthermore, many types of Gram-negative bacteria, for example E. coli and P. aeruginosa, synthesize isoprenoids via an alternative metabolic pathway (2C-methyl-D-erythritol 4-phosphate [MEP]), which do not require HMG-CoA reductase (Heuston et al., 2012). Yet, certain statins (ATV, RSV, and SMV) exert some antibacterial activity against E. coli, P. aeruginosa, and various Gram-negative bacteria (Table 2), likely via a mechanism independent of bacterial HMG-CoA reductase inhibition.

Knowledge gap: mechanism of statins’ antibacterial action (Postulated mechanism derived from structure-activity relationship analysis)

The mechanism of action for statins’ antibacterial effects has yet to be elucidated. The nature of antibacterial activity for SMV against Gram-positive bacteria was found to be bacteriostatic at drug concentrations that equal MIC (Thangamani et al., 2015), but bactericidal at concentrations four times greater than MIC (Graziano et al., 2015). Suggested mechanisms for statins’ antibacterial effects include the pleiotropic effects of statins repressing cell growth (Masadeh et al., 2012), or the hydrophobic nature of SMV disrupting bacterial membrane in a “soap-like” manner (Bergman et al., 2011), or the reduction of biofilm viability and production (Graziano et al., 2015). It was also hypothesized that by lowering host cholesterol levels, statins may reduce the production of a protective membrane-stabilising metabolite in the mevalonate pathway, resulting in bacterial cell toxicity (Haeri et al., 2015).

By comparing the chemical structures of statins with known antibacterial activity against statins without antibacterial activity, the presence of two methyl groups arranged in a tetrahedral molecular geometry were identified as important moieties responsible for statins’ antibacterial activity (Fig. 3). We postulate that statins may interfere with bacterial cell regulatory functions through non-polar interactions of statins’ methyl groups with alanine residues present in Gram-positive bacterial surface structures such as wall teichoic acids and lipoteichoic acids; hydrogen bond disruptions within Gram-negative bacterial surface lipopolysaccharide structures; and/or via hydrogen bonds and van der Waals forces with various other Gram-positive and Gram-negative bacterial surface proteins to exert bacteriostatic effects (or bactericidal effects at higher statin concentrations). The binding interactions may be similar to the manner by which antimicrobial peptides accumulate at bacterial surfaces (Malanovic & Lohner, 2016).

In Fig. 3, carbon atoms attached to two methyl groups arranged in a tetrahedral molecular geometry appeared to be common amongst the chemical structures of statins with antibacterial activity (SMV, ATV, FLV, and RSV). In particular, the structures of SMV and LVS are almost identical, except that SMV contains a carbon with two methyl groups in the ester side chain whereas LVS contains a carbon with only one methyl group. Since SMV has antibacterial effects against MRSA while LVS does not (Thangamani et al., 2015), this suggests the importance of the additional methyl moiety in the mechanism of action.

Bacteria have a high affinity for attaching to environmental surfaces, and one of the attachment methods involves non-polar interactions between a hydrophobic methyl group and a hydrophobic side group of an alanine residue (Boland, Latour & Stutzenberger, 2000). Repeating alanine residues are found in wall teichoic acids and lipoteichoic acids (Lebeer, Vanderleyden & Keersmaecker, 2010), which are important anionic polymers protecting bacteria against noxious environmental stress, assisting in bacteria colonisation, infection, and immune evasion (Brown, Santa Maria Jr & Walker, 2013; Xia, Kohler & Peschel, 2010). The two methyl groups from statins may be in the exact conformation (tetrahedral geometry) to directly bind with alanine residues of wall teichoic acids and lipoteichoic acids protruding from the peptidoglycan cell wall in Gram-positive bacteria (Silhavy, Kahne & Walker, 2010), causing structural distortions which may interfere with cell division (Hanson & Neely, 2012). In further support, an omission or decline in alanine residues of wall teichoic acids reduces biofilm adhesion and formation, as well as increases bacterial susceptibility to antibiotics, cationic antimicrobial peptides, phagocytes, and neutrophils (Brown, Santa Maria Jr & Walker, 2013).

There are also other surface proteins responsible for various roles in S. aureus such as adhering to and invading host cells, evading host immune responses, and formation of biofilms (Foster et al., 2014). Statins are able to change their conformation and bind extensively to proteins (≥88% protein binding, except for PRV which exhibits about 43% to 54% protein binding) through van der Waals forces and hydrogen bonds (Gazzerro et al., 2012; Shi et al., 2016). Therefore, the binding of statins to bacterial surface proteins may influence various metabolic pathways to reduce bacteria proliferation and virulence. This may account for the lack of antibacterial activity of PRV, which possessed significantly lower protein binding properties. Incidentally, amitriptyline (antidepressant), chlorpromazine (antipsychotic), propranolol (antihypertensive), and tamoxifen (anticancer) are other non-antibiotic drugs from different pharmacological classes which are highly protein bound (>90%), possess atoms attached to two methyl groups with a tetrahedral or a similar trigonal pyramidal molecular geometry, and also exhibit antibacterial activity against S. aureus (Fig. 3) (Kruszewska, Zareba & Tyski, 2004; Kruszewska, Zareba & Tyski, 2006; Kruszewska, Zareba & Tyski, 2010; Mandal et al., 2010).

The postulated mechanism of statins binding to bacterial cell surface structures and/or surface proteins also aligned with the results of two studies showing MIC[statin] (MRSA) >  MIC[statin] (MSSA) (Jerwood & Cohen, 2008; Masadeh et al., 2012). MRSA cocci are smaller than MSSA cocci and have a statistically higher cell surface to plasma volume ratio (Kocsis et al., 2010). As such, more statin drug may be required to bind to the corresponding higher number of surface attachments or proteins in MRSA, compared to MSSA cocci.

Gram-negative bacteria cells contain various exposed structures such as lipopolysaccharides and surface proteins protruding from the outer cell membrane (Lebeer, Vanderleyden & Keersmaecker, 2010). Lipopolysaccharide structures serve as a protective barrier and regulator of solutes (Rosenfeld & Shai, 2006; Ruiz, Kahne & Silhavy, 2009). Disruption of the stabilized hydrogen bond interactions within lateral lipopolysaccharide structures results in a possible breach in the barrier function (Ruiz, Kahne & Silhavy, 2009). Statins may bind to immobilized artificial membranes (which mimic the fluid phospholipid bilayer of cell membranes) via van der Waals forces and hydrogen bonds (Sarr, Andre & Guillaume, 2008). Hence some of the antibacterial effects exerted by statins on Gram-negative bacteria may be a result of statins’ hydrogen bond forces disrupting the lipopolysaccharide structure, and/or binding to the cell membrane surface proteins.

It was hypothesized that the inhibition of statins via the mevalonate pathway reduces a protective metabolite because the addition of cholesterol to Gram-positive (S. aureus and E. faecalis) and Gram-negative (E. coli and P. aeruginosa) bacteria decreased the antibacterial effects of statins (Haeri et al., 2015). The decreased antibacterial effect may be in part due to an increase in bacterial load as the in vitro addition of cholesterol has been shown to increase S. aureus growth (Shine, Silvany & McCulley, 1993). However, bacteria such as S. aureus and E. coli are able to incorporate exogenous cholesterol into their cell membranes (Eaton et al., 1981; Shine, Silvany & McCulley, 1993), increasing rigidity of the membranes and likely reduce disruptions of cell surface structures (Brender, McHenry & Ramamoorthy, 2012). Thus, statins may be unable to bind to rigid membranes in the required conformation, or are unable to distort cell surface structures, further supporting this review’s postulated mechanism of statins’ antibacterial activity.

More studies are required to accurately determine statins’ mechanism of antibacterial effects because if the antibacterial mechanism directly threatens bacteria survival, resistance develops more rapidly (Park & Liu, 2012). Even if statins are not repurposed as novel adjuvant antibiotics, their current extensive use for cardiovascular protection may still significantly influence susceptible bacteria.

AMR breaker: synergistic antibiotic effects

The combination of antibiotics with drugs that possess direct antibacterial properties or synergistic activity may impede AMR (Brown, 2015), especially when local delivery of drugs with different mechanisms of action are utilized (Brooks & Brooks, 2014). SMV exerted synergistic antibacterial effects against S. aureus clinical isolates with the topical antibiotics daptomycin, fusidic acid, mupirocin, and retapamulin (Thangamani et al., 2015). However, no synergism was found when SMV was combined with vancomycin against S. aureus (Graziano et al., 2015); when ATV, FLV, LVS, PRV, and SMV were each combined with amikacin, imipenem, or minocycline against A. baumannii (Farmer et al., 2013); or when ATV and FLV were each combined with ciprofloxacin, cefepime, or piperacillin-tazobactam against E. coli, K. pneumoniae, and P. aeruginosa respectively (Farmer et al., 2013).

AMR breaker: attenuated virulence factors

Virulence factors enable bacteria to harm the host (via adhesion, invasion, colonisation, and toxin secretion); or protect bacteria from the host’s immune defences (via secretion of immune response inhibitors, formation of capsules, and biofilms) (Wu, Wang & Jennings, 2008). Instead of directly threatening bacterial survival with antibiotics that affect essential bacterial genes, it has been suggested that non-threatening approaches such as disarming bacteria by attenuating virulence factors may help reduce AMR (Park & Liu, 2012).

Through the inhibition of Rho signaling activities and reduced cholesterol production, statins have been observed to attenuate virulence factors. Some examples include reducing bacteria motility and attachment, suppressing production of toxins (Panton-Valentine leucocidin and alpha-hemolysin), directly reducing bacterial translocation and invasion, or protecting against bacterial invasion indirectly via inhibiting lipid raft formation (Hennessy et al., 2016). Statins may also prevent biofilm formation, limit biofilm production, and reduce cell viability in matured biofilms (Graziano et al., 2015).

AMR breaker: enhanced host immunity

Stimulation of the host’s defence mechanisms to help resolve infections may potentially break AMR (Brown, 2015; Park & Liu, 2012). Statins have been shown to directly improve the host’s immune defence in humans as well as in animal models (Chow et al., 2010; Frostegard et al., 2016; Parihar et al., 2016; Walton et al., 2016; Yang et al., 2014). In humans, ATV and SMV may inhibit pro-inflammatory T cells and induce anti-inflammatory T regulatory cells via a novel method involving the downregulation of microRNA let-7c (Frostegard et al., 2016). Clinical studies revealed that SMV enhanced neutrophil function and improved chronic obstructive pulmonary diseases (Walton et al., 2016). In addition, women taking statins were less likely to be hospitalized due to the activation of lung macrophage nitric oxide synthase-3, which increases bacterial killing, clearance, and host survival in pneumonia (Yang et al., 2014). In animal models, SMV was found to protect mice against Leishmania major via augmented phagosome maturation and increased levels of oxidative hydrogen peroxide (Parihar et al., 2016).

However, statins may also unpredictably influence host immunity via factors such as NET production, pleiotropic effects during sepsis, and binding as agonists to nuclear receptors as discussed below. More studies are required in these ambiguous areas to determine the overall effects of statins on host immunity and consequently, whether statins potentially break or contribute to AMR.

Knowledge gap: neutrophil extracellular trap (NET) production

FLV, LVS, and SMV have been shown to produce NETs, which are complexes of nuclear DNA, histones, antimicrobial peptides, and proteases capable of trapping and killing a wide spectrum of microorganisms (Chow et al., 2010). However, there is also conflicting evidence that statins do not affect NET production (Sorensen & Borregaard, 2016). Further studies may be required to confirm the effect of statins on NETs, as well as whether the NET complexes are in sufficient concentrations to be antibacterial (Sorensen & Borregaard, 2016).

Knowledge gap: pleiotropic effects in sepsis

Statins may potentially benefit sepsis by reducing inflammation via intracellular signaling (Terblanche et al., 2007), lowering catecholamine levels (Millar & Floras, 2014), or reducing Toll-like receptor activation by pathogen associated molecular patterns (PAMPs) (Wittebole, Castanares-Zapatero & Laterre, 2010). Statins also possess antiangiogenic (at high doses) and antioxidant effects (Gazzerro et al., 2012), which may prevent the progression of severe sepsis (Vera et al., 2015). However, sepsis is a complex condition and there have been conflicting results of statins’ effects from meta-analysis studies (Bjorkhem-Bergman et al., 2010; Deshpande, Pasupuleti & Rothberg, 2015; Janda et al., 2010; Quinn et al., 2016).

During early sepsis, high levels of catecholamines and PAMPs such as lipopolysaccharides and lipoteichoic acids cause an initial pro-inflammatory response (Murphy et al., 2004; Rittirsch, Flierl & Ward, 2008). An anti-inflammatory response may be initiated concurrent to the initial inflammation and in some cases, secondary infections may cause a secondary pro-inflammatory response (Murphy et al., 2004). As sepsis continues, pathogenic bacteria may induce vagal stimulation to decrease catecholamines and suppress the host’s immune system (Weinstein, Revuelta & Pando, 2015). There are also many other pro-inflammatory factors (protein catabolism, cachexia, and persistent inflammation) and anti-inflammatory factors (defects in adaptive immunity) that occur slightly later after the onset of sepsis (Binkowska, Michalak & Slotwinski, 2015). These variables make it difficult to appropriately administer statins to reduce inflammation or catecholamine levels because it is uncertain if the host is in an overall state of immunostimulation or immunosuppression at any one point in time during sepsis.

Furthermore, the possibility of using statins in infections is further complicated by the potency of statins, whereby different types and doses of statins resulted in different outcomes (Ou et al., 2014). At low doses, statins exhibit proangiogenic effects (Gazzerro et al., 2012), which may be detrimental in severe sepsis (Vera et al., 2015). Hence varying administration times, different types or doses of statin could have caused the conflicting results in meta-analysis studies.

Knowledge gap: nuclear receptor agonists

Statins may indirectly influence the human immune system by binding as agonists to various nuclear receptors, namely farnesoid X receptors (FXRs), glucocorticoid receptors (GCRs), pregnane X receptors (PXRs), and vitamin D receptors (VDRs) (Howe et al., 2011; Marshall, 2006). Statins may also indirectly induce peroxisome proliferator-activated receptor gamma (PPARγ) activity (Paumelle & Staels, 2007). The activation of FXRs and VDRs induce antimicrobial peptide gene expression (Schaap, Trauner & Jansen, 2014), whilst activation of GCRs, PXRs, and PPARγ result in anti-inflammatory effects (Kadmiel & Cidlowski, 2013; Paumelle & Staels, 2007; Schaap, Trauner & Jansen, 2014).

Although statins may bind as agonists to nuclear receptors, a direct increase in nuclear receptor activity may not be apparent because by inhibiting the mevalonate pathway, statins reduce the production of several nuclear receptor agonists such as cholesterol (precursor of glucocorticoids which are GCR and PXR agonists), bile acids (FXR agonist), and vitamin D (VDR agonist) (Liao, 2005). Moreover, nuclear receptors may also influence the production of other receptor agonists (e.g., activation of PXR reduces bile acid production) (Schaap, Trauner & Jansen, 2014), and nuclear receptor agonists are not receptor specific (e.g., bile acids are agonists at both FXRs and VDRs; vitamin D is an agonist at GCRs, PXRs, and VDRs) (Gombart, 2009; Mangin, Sinha & Fincher, 2014; Marshall, 2006).

Some nuclear receptor agonists which boost the human immune system may ironically influence bacterial morphology directly to cause antibiotic tolerance (e.g., bile acids may activate FXRs and VDRs to stimulate antimicrobial peptide production, but bile acids also induce biofilm changes resulting in antibiotic resistant chronic infections) (Reen et al., 2016; Schaap, Trauner & Jansen, 2014). In view of the numerous variables, of which some are antagonistic, it is difficult to anticipate the net effect of statins on the immune system via nuclear receptor activity.

AMR breaker: improved wound healing

Uncomplicated skin and wound infections are amongst one of the highest causes for outpatient antibiotic usage (Hurley et al., 2013). As a result, inappropriate or prolonged antibiotic use may contribute to AMR. Antibacterial agents aiding in wound healing should serve to reduce bacterial infection and improve healing time, thus limiting exposure time to antibiotics. Statins are theoretically ideal for wound healing because they may act as PXR agonists to enhance wound healing in intestinal epithelial cells, inhibit FPP (an activator of GCR which impedes wound healing), reduce inflammation, regulate epithelial homeostasis, promote angiogenesis at low doses, reduce oxidative stress, increase vascular endothelial growth factors, and increase levels of nitric oxide (Bu, Griffin & Lichtman, 2011; Calanni et al., 2014; Elewa et al., 2010; Farsaei, Khalili & Farboud, 2012; Fitzmaurice et al., 2014; Vukelic et al., 2010). The effects of oral statins (ATV, SMV, LVS, PRV, and RSV) and topical statins (ATV, SMV, and LVS) have been examined and it was concluded that there was sufficient evidence to warrant clinical trials assessing the potential efficacy of statins in postoperative wound healing (Fitzmaurice et al., 2014).

AMR maker: dysbiosis of gut microbiota

Antimicrobials disrupting the gut microbiota may cause AMR and potentially create a store of AMR genes in the gut microbiota, resulting in recalcitrant infections (Francino, 2016). Statins have been shown to reduce gut microbiota diversity in humans (Zhernakova et al., 2016), but the mechanism of dysbiosis of the human gut microbiota has not been elucidated. A recent animal study has shown that statin-induced bile acid alterations resulted in mouse gut dysbiosis via a PXR-dependent mechanism (Caparros-Martin et al., 2017). Our review provides plausible evidence that statins may additionally disrupt the human gut microbiota via a direct antimicrobial effect.

From Tables 1 and 2, Gram-positive (E. faecalis, E. faecium, L. casei, and S. aureus) and Gram-negative (C. freundii, E. aerogenes, E. cloacae, E. coli, K. pneumoniae, and P. mirabilis) gut microbiota were susceptible to various statins, whereby MIC[SMV ] ≈ 8 to >500 µg/mL (Matzneller, Manafi & Zeitlinger, 2011; Ting, Whitaker & Albandar, 2016), MIC[ATV ] ≈ 16 to >1,024 µg/mL (Masadeh et al., 2012; Thangamani et al., 2015), MIC[RSV ] ≈ 100 to >1,024 µg/mL (Thangamani et al., 2015; Welsh, Kruger & Faoagali, 2009), and MIC[FLV ] ranged from >200 to >1,024 µg/mL (Jerwood & Cohen, 2008; Thangamani et al., 2015).

The licensed oral daily dose range of statins for cholesterol-lowering purposes are SMV = ATV = 10 mg to 80 mg (10,000 µg to 80,000 µg), FLV = 40 mg to 80 mg (40,000 µg to 80,000 µg), and RSV = 5 mg to 40 mg (5,000 µg to 40,000 µg) (Armitage, 2007). The laboratory conditions (35 °C and pH 7.2 to 7.4) at which MIC values were determined are attainable when gut microbiota are exposed to statins along the gastrointestinal tract (37 °C body temperature and pH 7.2 to 7.4 along various parts of the small intestines) (Clinical and Laboratory Standards Institute, 2012; Khutoryanskiy, 2015). Although gut concentrations of orally administered parent statin drugs are reduced via absorption, distribution, and metabolism as they move along the gastrointestinal tract, the reduction in concentrations are limited by enterohepatic circulation, and statins are eventually excreted mainly in the feces (SMV ≈ 60%, ATV > 98%, FLV ≈ 93%, and RSV ≈ 90%) (McFarland et al., 2014; Reinoso et al., 2002). As such, statin concentrations along the gastrointestinal tract are likely sufficient to kill gut microbiota. Even if gut statin concentrations fall below MIC, prolonged gut microbiota exposure to drug concentrations up to several hundred times lower than MIC may still result in selective pressures for resistance (Andersson & Hughes, 2011).

AMR maker: statin plasma concentrations in bacteremic patients being much lower than MIC

Oral doses of statins may be high enough to exert antimicrobial effects in the gut, but the peak statin plasma concentrations have been found to be much lower (SMV ≈ 0.0209 µg/mL, ATV ≈ 0.01 µg/mL, RSV ≈0.037 µg/mL, and FLV ≈ 0.24 µg/mL) due to low bioavailability and protein binding (Jerwood & Cohen, 2008; Kantola et al., 2000; Welsh, Kruger & Faoagali, 2009). Hence, statins are unlikely to exert significant systemic antimicrobial effects since the peak plasma concentrations range from hundred to thousand times lower than the MIC. Of greater concern however, is the risk of exposing bacteremic patients to such low systemic antimicrobial concentrations, which may result in selective pressures for resistance (Andersson & Hughes, 2011).

AMR maker: environmental impact due to extensive use of stains

The present usage of statins (ATV, RSV, and SMV) has resulted in residual levels (µg/mL to pg/mL) persisting in sewage for at least a few weeks (Lee et al., 2009; Ottmar, Colosi & Smith, 2012). Since the exposure of bacteria to antibiotic concentrations several hundred times below MIC (in the range of µg/mL to pg/mL) poses a risk of bacterial resistance (Andersson & Hughes, 2011), this lingering exposure of bacteria in the sewage system to current statin concentrations may thus contribute to selective pressures for resistance.

Conclusion

The potential roles of statins as AMR breakers, AMR makers, and knowledge gaps in the statin-bacteria-human-environment continuum have been summarized in Fig. 2. Literature has shown that SMV, ATV, RSV, and FLV exert varying antibacterial effects on Gram-positive and Gram-negative bacteria (Tables 1 and 2), especially SMV (against most of the Gram-positive bacteria tested) and ATV (against most of the Gram-negative bacteria tested). However, SMV currently appears to be the best candidate as a novel adjuvant antibiotic because it has been the most widely studied statin and demonstrated direct in vitro antibacterial activity against various types of microbiota (oral, gut, and nasopharyngeal), drug-resistant bacteria, and environmental bacteria. Based on the structure-activity relationship analysis of statins’ chemical structures, it is plausible that statins’ mechanism of antibacterial activity involves the interference of bacterial cell regulatory functions via binding to bacterial cell surface structures such as wall teichoic acids and lipoteichoic acids (for Gram-postive bacteria), lipopolysaccharides (for Gram-negative bacteria), and/or bacterial surface proteins (for both Gram-positive and Gram-negative bacteria).

Current evidence better supports statins as AMR breakers by working synergistically with existing topical antibiotics, attenuating virulence factors, boosting human immunity, or aiding in wound healing. However, the paucity of data directly associating statins to AMR should not exclude statins’ role as plausible AMR makers. The widespread use of statins for non-antibiotic (cardioprotective) purposes may favor selective pressures or co-selection for resistance via dysbiosis of the human gut microbiota, sublethal plasma concentrations in bacteremic patients, and persistence in the environment, all of which could culminate in AMR.

Perhaps the most urgent knowledge gap to address is determining the mechanism of statins’ antibacterial activity. If the antibacterial mechanism involves disarming bacteria instead of directly threatening bacterial survival, AMR is not likely to develop rapidly (Park & Liu, 2012), and statins may still play an effective role as AMR breakers. However, if the antibacterial mechanism directly threatens bacterial survival, AMR is likely to develop rapidly. If so, statins’ role as AMR breakers will likely be limited, and may paradoxically function as AMR makers instead.

Supplemental Information

Supplemental Information 1 PRISMA 2009 Checklist for Statins: Antimicrobial resistanc breakers or makers?”

Click here for additional data file.

The authors wish to thank all their friends and colleagues at the Curtin Health Innovation Research Institute (CHIRI) Biosciences Research Precinct Core Facility (Curtin University), especially Dr. Joshua Ramsay, for making this work possible.

Additional Information and Declarations

Competing Interests

Author Contributions

Data Availability

The authors declare there are no competing interests.

Humphrey H.T. Ko wrote the paper, prepared figures and tables, reviewed drafts of the paper, designed the review, performed the literature search, selected relevant references, analysed the references and contributed ideas to the review.

Ricky R. Lareu and Brett R. Dix reviewed drafts of the paper, analysed the references and contributed ideas to the review.

Jeffery D. Hughes reviewed drafts of the paper, conceived and designed the review, analysed the references and contributed ideas to the review.

The following information was supplied regarding data availability:

The raw data is included in Tables 1 and 2.

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
