# Peer review of "Statins: antimicrobial resistance breakers or makers?"

_PeerJ, doi:10.7717/peerj.3952_

## Round 0.1 · original submission · Major Revisions

· Academic Editor

Major Revisions

Based on the comments provided by reviewers, I think it would be highly convenient to focus and better structure this manuscript, particularly by paying attention to concerns raised by reviewer 2. Should you address all these comments, as well as the minor ones raised by reviewer 1, I will be happy to receive your revised version.

·

Basic reporting

No comment

Experimental design

No comment

Validity of the findings

No comment

Additional comments

I consider that the review is well-written and well-structured. It provides new information related to the antibacterial effects of statins, which results useful for the control of bacterial infections that can become in resistant strains. The review gives information about the evidences that deal with the development of bacterial resistance in response to statins, and those that agree with the opposite effect (“AMR breakers”) of these molecules.

In my opinion, this review is worth to be published in PeerJ. I only have minor concerns that should be corrected in the manuscript:

1. In line 284, authors describe that the mechanism for the antibacterial effects of statins could be apoptosis. Please, eliminate this sentence, since apoptosis is a cell death mechanism restricted to eukaryotic cells. The term “apoptosis” to explain regulated cell death in bacteria is under discussion, and its molecular basis are very different from that detected in eukaryotic cells. For your reference, Microbes Infect. 2013 Jul-Aug;15(8-9):640-4. doi: 10.1016/j.micinf.2013.05.005.

2. The period (.) before each reference is wrong. It should be located after.

3. The sentence in Lines 285-288 is confusing, and it appears contradictory: How the reduction in a metabolite in the mevalonate pathway, essential for bacterial membrane stability, can protect bacteria from statins’ cytotoxic effects? It should be the opposite effect. Please, clarify.

4. Is there any study related to the analysis of microbiota in people under prolonged statin treatment? This kind of information is missing in the review, and will strength the discussion related to the negative effects of statin against microbiota.

Reviewer 2 ·

Basic reporting

This work presents a compilation of MIC data for Statins against Gram-positive and Gram-negative bacteria (Results and Tables 1 and 2), proposed mechanism(s) of action of Statins, and knowledge gaps in the field.

The manuscript is well written. References, background and tables are provided in sufficient detail.

Line 235 – Title needs editing. (suggestion: Fungal origin does not correlate with antibacterial activity/efficacy.)

Experimental design

Although a summary of the current status of research on Statins may be useful for researchers in the field, the specific aims and scope of this review are unclear.

In the Introduction (Conclusion), it is claimed that this review 'provides evidence' of a link between statins and antimicrobial resistance. This is inaccurate, as the only data presented in this work are MIC values from previous studies and molecular structures of Statins. At best, this manuscript proposes how synergistic use of statins with other antibiotics may affect microbial outcomes.

As detailed in section 3 below (validity of the findings), the utility of summarizing the MICs is not obvious, especially in the light of significant variation in MIC data and the limited scope of MIC assays as tools for understanding mechanism of antibacterial action.

It is suggested that whether statins support or oppose primary drugs in their killing action is contextual. Several examples are presented in the discussion section, based on previously published works. Although this is a significantly important part of the review, it seems disconnected from the previous analysis of MICs and plausible mechanism.

Validity of the findings

The Results section summarizes MIC values of various statins reported in the previously published studies. The authors then go on to list the potential pitfalls of MIC analysis, including differences in assay times, growth media and other experimental conditions (e.g. with or without shaking). Given the large variation (~5-fold) in MIC values reported for the same statin with the same bacterium, the utility of the MIC comparisons made by the authors seems limited. It is important to note that MIC assays have a limited scope as far as elucidating the mechanism of antimicrobial action is concerned. Within these limitations, the authors have analyzed the structures of Statins and proposed a plausible mechanism of action, namely binding to the bacterial outer membrane proteins, wall teichoic acids and lipoteichoic acids. It is not clear how the authors arrived at this based on their analysis of MICs and/or the structure of Statin molecules.

I urge the authors to clarify how the quarternary chiral carbon, and attached methyl groups, might affect the mechanism of action? Suggestion of a possible interaction between the methyl groups and alanine groups of teichoic acids is speculative. Is there data on Statin-bacteria interactions to support this statement? For e.g. are there any docking studies of the di-methyl moiety to components of LPS, lipid bilayer, outer membrane proteins, etc?

Lines 55-59: Difference in the MIC and sub-MIC regimes is not clear. Both seem to involve growth of resistant bacteria and inhibition of growth of susceptible bacteria.

Lines 103 -106 are identical to lines 15-18.

Line 113 – ‘relative’ and ‘comparison’ are redundant. ‘quantitative’ comparison might be more appropriate. If the authors agree, this change should be made for all instances in the text.

Line 251 – Absence of antimicrobial activity of PRV could be due to other factors (for e.g. absence of quarternary chiral carbon, which the authors have suggested might be important for binding to outer membrane components). These may be essential or preceding steps in the action of statins, prior to involvement of the dihydroxy acid moiety. Therefore, inhibition of HMG-CoA could still be one of the mechanisms of action of statins.

Line 321 – Binding to statins to proteins is suggested as one of the mechanisms of the inhibitory action of Statins. The readers might benefit from a table summarizing the % binding for various Statins.

Lines 326-336: The hypothesis that more membrane area per volume requires more statin molecules seems an oversimplification. There could be other differences between MRSA and MSSA that may explain the difference in MIC values. For e.g. It is unclear how the virulence of both strains compares. If MRSA were more virulent than MSSA, that would significantly affect the MIC values.

Line 304 – Please clarify what is meant by ‘bacterial adhesion’.

Line 309 – Suggestion of a possible interaction between the methyl groups on Statins and alanine groups of teichoic acids is speculative. Is there data on Statin-bacteria interactions to support this statement?

Line 321 – Binding to statins to proteins is suggested as one of the mechanisms of the inhibitory action of Statins. The readers might benefit from a table summarizing the % binding for various Statins.

Line 340-347 – Statins are electrically neutral. How do they interact with the bacterial LPS (hydrophilic) which is charged, and the phospholipid bilayer (anionic)?

Line 547-550 – It is not clear what the authors are trying to convey here.

Additional comments

I suggest the authors re-analyze the purpose and scope of this manuscript and revise this draft prior to resubmission. The purpose of the MIC analysis needs to be considered and clarified in any revisions the authors might make. A possible route to redrafting this manuscript would be to focus solely on the synergistic action of Statins with other antibiotics. This might help provide a clear framework for the review, and serve as an important resource for researchers interested in the effects of synergistic drug treatment on antimicrobial resistance.

---

## Round 0.2 · accepted · Accept

· Academic Editor

Accept

I can see that you throughly modified your work, with a more balanced structure and addressed concerns regarding the use of MIC data, and also those needed for grammar and style. I will look forward to receiving your new submissions to PeerJ.

·

Basic reporting

No comment

Experimental design

No comment

Validity of the findings

No comment

Additional comments

I consider that the manuscript has been significantly improved including all the suggestions that I made at the first round of the peer review process.